# Potentiation of NMDA Receptors by AT1 Angiotensin Receptor Activation in Layer V Pyramidal Neurons of the Rat Prefrontal Cortex [note 1]

**DOI:** 10.3390/ijms252312644

**Published:** 2024-11-25

**Authors:** Adrienn Hanuska, Polett Ribiczey, Erzsébet Kató, Zsolt Tamás Papp, Zoltán V. Varga, Zoltán Giricz, Zsuzsanna E. Tóth, Katalin Könczöl, Ákos Zsembery, Tibor Zelles, Laszlo G. Harsing, László Köles

**Affiliations:** 1Department of Oral Biology, Semmelweis University, H-1089 Budapest, Hungary; hanuska.adrienn@dpckorhaz.hu (A.H.); ribiczey.polett@gmail.com (P.R.); papp.zsolt@semmelweis.hu (Z.T.P.); zsembery.akos@semmelweis.hu (Á.Z.); zelles.tibor@semmelweis.hu (T.Z.); 2Department of Pharmacology and Pharmacotherapy, Semmelweis University, H-1089 Budapest, Hungary; kato.erzsebet@semmelweis.hu (E.K.); varga.zoltan@semmelweis.hu (Z.V.V.); giricz.zoltan@semmelweis.hu (Z.G.); harsing.laszlo@semmelweis.hu (L.G.H.J.); 3Laboratory of Neuroendocrinology and In Situ Hybridization, Department of Anatomy, Histology and Embryology, Semmelweis University, H-1094 Budapest, Hungary; toth.zsuzsanna.emese@semmelweis.hu (Z.E.T.); konczol.katalin@semmelweis.hu (K.K.); 4Laboratory of Molecular Pharmacology, HUN-REN Institute of Experimental Medicine, H-1083 Budapest, Hungary

**Keywords:** NMDA receptor, AT_1_ angiotensin receptor, D1 dopamine receptor, prefrontal cortex, RAS, neuromodulation

## Abstract

NMDA receptors in the prefrontal cortex (PFC) play a crucial role in cognitive functions. Previous research has indicated that angiotensin II (Ang II) affects learning and memory. This study aimed to examine how Ang II impacts NMDA receptor activity in layer V pyramidal cells of the rat PFC. Whole-cell patch-clamp experiments were performed in pyramidal cells in brain slices of 9–12-day-old rats. NMDA (30 μM) induced inward currents. Ang II (0.001–1 µM) significantly enhanced NMDA currents in about 40% of pyramidal cells. This enhancement was reversed by the AT_1_ antagonist eprosartan (1 µM), but not by the AT_2_ receptor antagonist PD 123319 (5 μM). When pyramidal neurons were synaptically isolated, the increase in NMDA currents due to Ang II was eliminated. Additionally, the dopamine D1 receptor antagonist SCH 23390 (10 μM) reversed the Ang II-induced enhancement, whereas the D2 receptor antagonist sulpiride (20 μM) had no effect. The potentiation of NMDA currents in a subpopulation of layer V pyramidal neurons by Ang II, involving AT1 receptor activation and dopaminergic signaling, may serve as an underlying mechanism for the effects of the renin–angiotensin system (RAS) elements on neuronal functions.

## 1. Introduction

Neuronal activity in the mesoprefrontal cortex (mPFC) plays a central role in human cognition, the execution of goal-oriented behaviors, and decision-making processes [1,2,3]. The PFC’s connections with other cortical and subcortical regions of the brain are essential for advanced cognitive functions such as learning, problem-solving, and goal-directed behavior [4,5,6,7]. The PFC is crucial for filtering out distractions and selecting what is important for the task [1]. Dysfunctions within these complex systems contribute to pathological conditions such as cognitive impairments and schizophrenia [8,9,10,11]. A pivotal element of PFC function is the layer V/VI pyramidal neuron, which integrates incoming glutamatergic and dopaminergic inputs from the thalamus and ventral tegmental area, respectively, before relaying signals to various brain regions [12,13,14]. Alterations or dysregulations in the glutamatergic or the dopaminergic systems may impair PFC functions [11,15,16,17].

The systemic renin–angiotensin system (RAS) produces hormones crucial for cardiovascular control. Angiotensin II (Ang II) is also produced locally in tissues, including the brain, where it serves important functions [18,19,20,21]. Ang II is linked to brain processes such as cognition and memory [22]. The central nervous system (CNS) produces all essential precursors and enzymes necessary to form and process the biologically active forms of the RAS [23]. Ang II synthesis in the brain is generally thought to result from the cooperation of multiple cell types, including neurons and glial cells [24,25,26]. Astrocytes are widely regarded as the primary source of brain angiotensinogen [25,27,28,29], and they constitutively secrete angiotensinogen into the brain’s extracellular fluid [30]. Evidence also suggests the presence of angiotensinogen in neurons [26,31]. Renin mRNA has also been detected in the rodent brain, but its low expression levels have raised questions about its functional relevance. However, the high levels of prorenin and the prorenin receptor, which binds both renin and prorenin, potentially sequestering and activating them, may provide a viable mechanism for Ang II synthesis within the CNS [32,33,34,35]. The angiotensin-converting enzyme (ACE) is widely distributed in the brain, although only low levels have been reported in cortical areas [36]. An alternative possibility is that Ang II may be generated from angiotensinogen through non-ACE pathways, involving enzymes such as cathepsin G or chymase [27].

Ang I is inactive, whereas Ang II and Ang III act as agonists at the AT_1_ and AT_2_ receptor subtypes. The AT_1_ receptor mainly couples with G_q/11_ proteins, while the AT_2_ receptor is associated with G_i_ proteins [37,38,39,40]. Both AT_1_ and AT_2_ receptors are expressed in the brain. AT_1_ receptor expression is predominantly observed in the regulatory and sensory integration centers of the brainstem and hypothalamus, including the area postrema, nucleus tractus solitarii, and medial preoptic nucleus, as well as in the anterior pituitary gland, amygdala, piriform cortex, and lateral olfactory tract. AT_2_ receptor expression is notably high in certain motor nuclei of the medulla oblongata, such as the hypoglossal nucleus and inferior olivary nucleus, and also in the habenula, basal ganglia, locus coeruleus, midbrain colliculi, thalamus, amygdala, and ventral tegmental area. In comparison, their expression in cortical areas is relatively moderate [41,42,43,44,45,46]. Ang IV is the ligand at AT_4_ receptors, abundant in the brain, including cortical areas [47]. The identity of the AT_4_ receptor is disputed, possibly being the insulin-regulated aminopeptidase (IRAP) or the c-Met growth factor receptor [47,48,49]. Another G protein-coupled receptor, Mas, may specifically bind Ang1-7, found in various brain regions, including the cortex [50,51].

Glutamate receptors are categorized into ionotropic receptors, which include N-methyl-D-aspartate (NMDA) and alpha-amino-3-hydroxy-5-methylisoxazole-4-propionic acid (AMPA)/kainate types, and metabotropic receptors [52]. NMDA receptors are ubiquitously distributed across the brain, with high expression levels in the PFC [53,54]. These receptors are critical mediators of excitatory synaptic currents. Activation of NMDA receptors requires the simultaneous binding of glutamate and co-agonists (serine or glycine). These receptors are uniquely blocked by Mg^2^⁺, which is relieved upon depolarization, thereby functioning as coincidence detectors of presynaptic and postsynaptic neuronal activity. Upon activation, they permit Ca^2^⁺ influx, triggering signaling pathways that can induce short-term potentiation lasting a few hours. More substantial depolarization results in a greater influx of Ca^2^⁺, activating various signaling cascades, including those involving protein kinases. This leads to the phosphorylation of numerous proteins, resulting in both biochemical and structural changes within the neuron. These alterations contribute to long-lasting changes in neuronal activity, such as long-term potentiation (LTP), a form of synaptic plasticity that can last from several hours to months. LTP is considered a fundamental mechanism underlying learning and memory, representing one of the cellular bases of synaptic plasticity [55,56]. Conversely, excessive NMDA receptor activity leading to uncontrolled Ca^2+^ influx can result in cell death and neurodegeneration [55]. Excitatory transmission is highly sensitive to modulatory influences, with the overall effect determined by the interplay of various factors [57]. Consequently, studying potential novel modulatory effects on NMDA receptors in the PFC is of considerable (patho)physiological importance.

Given the presence of various angiotensin (Ang) receptors in memory-related areas, it is unsurprising that different angiotensins impact cognitive functions [19,21,22,58,59]. The RAS can affect cholinergic, adrenergic, and dopaminergic transmission, all crucial for higher-order cognitive processes [60,61,62,63]. The RAS may be linked to brain disorders such as anxiety, depression, neurodegenerative diseases, and epilepsy, with its components being potential therapeutic targets for these conditions [18,19,21,22,58,59,64,65]. Despite increasing knowledge of angiotensins’ effects on neuronal functions, there are no significant data available on their impact on the PFC, which is crucial for organizing higher-order cognitive functions. Information about how angiotensins might affect NMDA receptors in cortical regions is also limited. AT_1_ receptor activation inhibited neuronal firing rates, long-term potentiation (LTP), and learning and memory, while AT_4_ receptor activation facilitated these processes [66,67]. AT_2_ receptor activation attenuated NMDA receptor-mediated signaling in cultured neurons [68,69]. Ang II enhanced LTP in the hippocampus, suppressed NMDA-dependent long-term depression (LTD) in the amygdala [70,71,72], and reduced glutamate responses in the locus coeruleus [73,74]. To our knowledge, a specific interaction between Ang and NMDA receptors in the PFC has not yet been revealed. In this study, we investigated the effects of Ang II on NMDA receptor function in layer V pyramidal cells of the PFC using the patch-clamp technique.

## 2. Results

### 2.1. NMDA Induced Inward Currents in Layer V Pyramidal Neurons of the Rat PFC, Which Were Reproducible at Lower Concentrations

NMDA was administered at concentrations ranging from 10 to 300 μM to generate a concentration–response curve for the agonist (Figure 1). The agonist was administered at a holding potential of –70 mV in increasing concentrations over three 1.5 min sessions (T_1_, T_2_, and T_3_), each separated by 10 min superfusion periods with drug-free artificial cerebrospinal fluid (aCSF; Figure 2A). In a different set of experiments, NMDA was administered three times at the same concentrations; there was a marked reduction in current responses from T_1_ to T_2_ and a further decrease from T_2_ to T_3_ at concentrations of 100 μM NMDA or higher. Consequently, a submaximal concentration of 30 μM was chosen for the subsequent experiments. This concentration, corresponding to the lower range of the concentration–response curve (Figure 1), minimized the risk of excessive neuronal depolarization and spontaneous epileptic activity, which can occur with higher concentrations of glutamate receptor agonists. By employing this submaximal NMDA concentration, we aimed to selectively activate NMDA receptors without inducing widespread depolarization or triggering network-level hyperexcitability. In accordance with previous whole-cell patch-clamp studies in rat PFC pyramidal neurons, we observed a slight reduction in the current induced by NMDA (30 μM) from the first to the second administration (see Wirkner et al., 2002 and 2007 for further details) [75,76]. Therefore, we focused on evaluating T_2_ and T_3_. NMDA (30 μM) consistently elicited repeatable currents at T_2_ and T_3_, with a T_3_/T_2_ ratio of 104.16 ± 4.58% (Figure 2B and Figure 3A, left column).

### 2.2. Ang II Potentiated NMDA-Induced Inward Currents in Layer V Pyramidal Neurons of the Rat PFC

Ang II, at concentrations ranging from 1 to 1000 nM, enhanced the effect of NMDA (30 μM) when applied 5 min before and during T_3_ in a subpopulation of pyramidal cells (Figure 3). Specifically, pyramidal cells exhibited a bimodal response to Ang II: one population’s responses were statistically indistinguishable from the control, while the other population, comprising approximately 40% of the investigated cells, demonstrated a statistically significant potentiation. For further analysis, only the latter subgroup, which exhibited enhanced currents in response to Ang II administration, was considered. The exclusion criterion for potentiation was based on an arbitrary cutoff of 116%, calculated from the control mean, standard error of the mean (SEM), and 99% confidence interval. Neurons displaying responses below this threshold were excluded, as they did not meet the criteria for significant enhancement by Ang II.

Figure 3 displays the normalized current response to NMDA (%) in responsive cells at 1 nM (11/28 cells, 150.9 ± 8.66%, *p* < 0.05), at 10 nM (9/22 cells, 152.93 ± 11.58%, *p* < 0.05), and at 1000 nM Ang II concentrations (21/47 cells, 178.56 ± 14.73%, *p* < 0.05). Interestingly, the application of 10 μM Ang II led to a reduction in both the magnitude of potentiation in responsive cells (138.63 ± 9.32%) and the proportion of cells exhibiting potentiation (4 of 16 cells, 25%). The administration of micromolar concentrations of Ang II also resulted in the sporadic appearance of inhibitory responses in a subset of cells, approximately one out of every thirteen.

### 2.3. AT1 Receptors for Ang II Are Expressed in the Rat PFC, and the Ang II-Induced Potentiation Was Mediated by These Receptors

When the AT_1_ receptor antagonist eprosartan (1 μM) was superfused throughout the experiment, Ang II (1 μM) failed to potentiate the 30 μM NMDA-induced currents in any pyramidal cells (T_3_/T_2_, 99.31 ± 3.95%, *p* > 0.05, *n* = 8). Conversely, when PD 123329, an AT_2_ receptor blocker (5 μM), was present, Ang II (1 μM) enhanced the NMDA (30 μM)-induced currents in 5 out of 11 pyramidal neurons (T_3_/T_2_, 163.74 ± 20.45%, *p* < 0.05) (Figure 4A). Neither eprosartan (1 μM) nor PD 123329 (5 μM) had any effect on the NMDA-induced currents when administered separately (98.9 ± 3.16, *n* = 7, and 101.3 ± 3.57, *n* = 6, respectively).

Immunohistochemical staining with the MBS151548 anti-AT1R rabbit polyclonal antibody (left panel, Figure 4B) and fluorescent in situ hybridization for AT_1_ angiotensin receptor mRNA expression (right panel, Figure 4B) demonstrated the presence of AT_1_ receptors in the PFC of 10-day-old rat brains. These data show that the AT_1_ receptors are highly expressed in both layer V-VI and layer II-III cells, including pyramidal neurons.

### 2.4. Synaptic Isolation of Pyramidal Neurons, as Well as Simultaneous D1 Antagonism, Abolished the Enhancement of NMDA Currents by Ang II

When pyramidal cells were isolated from neighboring neurons using tetrodotoxin (TTX, 0.5 μM) or a Ca^2+^-free medium, Ang II (1 μM) failed to potentiate the NMDA (30 μM)-induced currents in any of the pyramidal cells (T_3_/T_2_, 91.25 ± 9.81, *n* = 8 and 91.32 ± 2.7, *n* = 15, for TTX and Ca^2+^-free solution, respectively; Figure 5). Similarly, when SCH 23390, a D1 receptor antagonist, was continuously applied at a concentration of 10 μM throughout the experiment, Ang II (1 μM) did not enhance the NMDA (30 μM)-induced currents in any of the pyramidal cells (T_3_/T_2_, 87.83 ± 11.64, *n* = 8). Conversely, when sulpiride (20 μM), a D2 dopamine receptor antagonist, was present, Ang II (1 μM) enhanced the NMDA (30 μM)-induced currents in 8 out of 20 pyramidal cells to an extent similar to that measured in the absence of the antagonist (T_3_/T_2_, 145.42 ± 11.94). Interestingly, in those groups where potentiation was not present, a slight tendency toward inhibition was observed due to the previously mentioned sporadic appearance of cells showing inhibitory responses (Figure 5, second, third, and fourth columns).

## 3. Discussion

The primary finding of this study is that Ang II enhances NMDA receptor responses in a subset of layer V pyramidal neurons within the prefrontal cortex, likely through a presynaptic interaction involving dopaminergic mechanisms. This potentiation occurred via the activation of AT_1_ angiotensin receptors, but not AT_2_ receptors, and was observable over a broad concentration range of Ang II, from 1 nanomolar to 10 micromolar. Immunohistochemical and fluorescent in situ hybridization analyses showed the expression of AT_1_ angiotensin II receptors in both layer V–VI and layer II–III cells, including pyramidal neurons of the PFC.

Our previous research has identified several modulatory influences on NMDA receptors within central dopaminergic regions involved in higher-order cognitive functions, such as the PFC, involving both pre- and postsynaptic mechanisms [77,78]. In addition to the D1 dopamine receptor-mediated potentiation of NMDA responses [79], adenosine triphosphate (ATP), a potential co-transmitter of dopamine, has been reported to influence glutamatergic excitation in the PFC. Specifically, P2Y receptors exhibited positive interaction with NMDA receptors on layer V pyramidal neurons, akin to dopamine. This involved a presynaptic mechanism: astrocytic P2Y_4_ receptor activation by ATP released vesicular glutamate onto neighboring neurons, which in turn stimulated type I metabotropic glutamate (mGlu) receptors, enhancing NMDA currents [75,76]. It was also revealed that while exocytotic glutamate release activates group I mGlu receptors, glutamate accumulation due to astrocytic uptake blockade stimulates group II mGlu receptors. Both groups of mGlu receptors interacted with NMDA receptors to facilitate their function [80]. However, our research group also reported an inhibitory interaction between P2Y and NMDA receptors mediated by the P2Y_1_ subtype of metabotropic ATP receptors. This interaction involves membrane-delimited crosstalk between P2Y_1_ and NMDA receptors as a proposed underlying mechanism [75,81].

Our current data, demonstrating AT_1_ receptor-mediated potentiation of NMDA responses, suggest the involvement of synaptic transmission, including presynaptic factors such as neurotransmitter release, potentially implicating the dopaminergic system. Two lines of indirect evidence support this hypothesis. Firstly, the effect of Ang II was abolished when pyramidal neurons were partially synaptically isolated using a Ca^2+^-free medium or tetrodotoxin. Secondly, blockade of D1 receptors (but not D2 receptors) prevented the potentiation induced by Ang II.

Dopaminergic neurotransmission significantly influences prefrontal cortical circuitry [82]. There is ample evidence demonstrating that dopamine enhances NMDA currents through activation of D1 receptors [79,83,84,85,86]. PFC neurons receive prominent dopaminergic inputs from the ventral tegmental area, and the basis for crosstalk between NMDA and D1 receptors is established by the close adjacency of glutamatergic and dopaminergic axon terminals on PFC pyramidal neurons [12,84]. Pyramidal neurons in the PFC express mRNA for all five subtypes of dopamine receptors, as well as mRNA for the NMDA receptor [52,54,87,88]. Here, we have shown significant expression of AT_1_ receptors in multiple cell types within the L2-3 and L5-6 layers of the rat PFC.

Ang II has been shown to elicit dopamine release in the brain, which can be blocked by AT_1_ receptor antagonists [89,90]. Interactions between dopamine and the RAS have been observed in various central and peripheral tissues, and perturbations in these interactions are implicated in pathological conditions. For example, in the nigrostriatal system, dysregulated local RAS and disrupted angiotensin–dopamine interactions may contribute to the initiation and progression of dopaminergic neuron degeneration in Parkinson’s disease [65]. The brain RAS likely influences memory and cognitive functions, and its disruption may contribute to the development or progression of Alzheimer’s disease [22]. Blocking AT_1_ receptors is suggested to be advantageous in neurodegenerative conditions such as Alzheimer’s and Parkinson’s diseases, traumatic brain injury, and radiation-induced brain damage, as well as stress and mood disorders [58,59,64].

While the inhibitory effect of SCH-23390 strongly suggests the involvement of D1 receptors in the Ang II-induced potentiation of NMDA currents, some caution is warranted in interpreting these data. Notably, SCH-23390, at micromolar concentrations, has also been reported to inhibit G protein-coupled inwardly rectifying potassium (GIRK) channels [91]. Activation of AT_1_ receptors can lead to both facilitation and inhibition of GIRK currents [92], with inhibition potentially resulting in membrane depolarization that may enhance NMDA receptor activity. However, in this context, the possible inhibition of GIRK channels by SCH-23390 is not expected to reverse the Ang II-mediated potentiation of NMDA receptor responses. Nevertheless, further investigation could provide insights into this possibility.

Although we applied a submaximal concentration of NMDA in our experiments and Ang II did not significantly alter the baseline during superfusion, we cannot rule out the involvement of AMPA and GABA receptors in the Ang II-mediated potentiation of NMDA currents, as these receptors were not blocked. AMPA receptors contribute to postsynaptic depolarization, which is essential for relieving the magnesium block of NMDA receptors. If Ang II enhances AMPA receptor activity, it could increase excitatory input, indirectly facilitating NMDA receptor activation. Ang II may also modulate AMPA receptor dynamics, potentially influencing synaptic plasticity mechanisms such as LTP, thereby amplifying NMDA receptor-mediated effects. In a network context, AMPA receptors, which mediate fast excitatory transmission, could influence the firing patterns of layer V pyramidal cells [55,56,57]. GABA receptors, which mediate inhibitory transmission, could also modulate the net depolarization of pyramidal neurons and contribute to the observed effect. GABAergic interneurons may regulate pyramidal cell activity, potentially altering NMDA receptor-mediated currents [93]. Thus, while the lack of baseline change suggests that a direct effect on postsynaptic AMPA receptors is unlikely, both AMPA and GABAergic inputs could still be influenced by Ang II and thus play a role in the overall neuronal response to Ang II.

The observation that only a subset of layer V pyramidal neurons is involved in the AT_1_ and D1 receptor-mediated interaction is not uncommon. Notably, two early studies demonstrated that Ang II induced excitation in 11.2% of layer V–VI neurons in the rat somatosensory cortex [94] and in 33% of large layer V pyramidal neurons in the rat primary motor cortex [95], respectively. Pyramidal neurons in the PFC exhibit significant heterogeneity, with their physiological properties varying widely [96]. Yang et al. identified four principal types of pyramidal cells in layers V and VI of the rat PFC, each with distinct electrophysiological properties [14]. D1 receptors have been reported to be present in only a specific subpopulation of layer V pyramidal neurons [97]. A more recent study revealed that D1 receptors are enriched in layer V and VI pyramidal cells that project to intratelencephalic, but not extratelencephalic, targets, and that these receptors selectively enhance action potential firing in a subset of neurons [98]. In this study, 33% of layer V and 48% of layer VI pyramidal cells in the prefrontal cortex of mice were positive for D1 receptors. In the present study, we investigated a mixed population of layer V–VI pyramidal neurons, likely including both D1 receptor-positive and D1 receptor-negative neurons. Therefore, our finding that NMDA receptors were potentiated by Ang II/dopamine in only a subset of pyramidal neurons is in accordance with the scientific literature. Furthermore, the proportion of Ang II-responsive cells is similar to the proportion reported to be D1 receptor-positive by Anastasiades et al. [98]. A similar phenomenon was observed in our previous research, which investigated purinergic modulation of excitatory glutamatergic transmission in a mixed population of pyramidal cells in the rat PFC. These findings suggest that the effects are not diffuse actions affecting most cells in the layer; rather, they reflect specific mechanisms influencing the activity of distinct neuronal populations. These studies, which reveal novel modulatory influences on glutamatergic transmission, lay the groundwork for future, more detailed analyses of these effects and their underlying mechanisms in specific cell types.

The degree of potentiation remained relatively constant across a broad concentration range of Ang II (1 nanomolar to 1 micromolar). Although concentration dependence is an important factor in validating specific mechanisms, the reversal of potentiation by an AT_1_ receptor antagonist strongly supports the specificity of the Ang II effect. This antagonist-sensitive response demonstrates that NMDA receptor potentiation is mediated by AT_1_ receptor activation, even in the absence of a graded dose–response effect over the tested concentration range. We hypothesize that the concentrations tested represent the upper part of the concentration–response curve, where the potentiation effect has reached a plateau. The available scientific data regarding the (patho)physiologically relevant concentrations of Ang II in the rat cerebral cortex are sparse; it is likely in the picomolar range [99,100]. In patch-clamp experiments using brain slices, the concentration of a substance in the superfusion solution is typically higher than its actual concentration in the tissues, especially in the microenvironment where the investigated receptors are located. Nevertheless, it is possible that the nanomolar to micromolar concentrations of angiotensin II applied in our experiments were supraphysiological, and that even lower concentrations may be (patho)physiologically relevant. Further investigation is required to confirm this hypothesis.

Several potential mechanisms for the proposed AT_1_–dopamine-mediated interaction may be envisioned, possibly involving various cell types. In this study, AT_1_ receptor expression was clearly observed in cells of layers V–VI and II–III in the PFC. However, further analysis to identify the exact origin and specific anatomical localization of the RAS components involved in these interactions was beyond the scope of this work. Angiotensinogen is primarily produced and constitutively secreted by astrocytes, but it is also synthesized by neurons, where it may either be secreted or retained intracellularly [101,102]. The paracrine pathway of Ang II generation, involving cooperation between glial cells and neurons, can lead to the activation of AT_1_ receptors on the surface of neurons. Astrocytes also express AT_1_ receptors on their cell surface. Additionally, an intracellular RAS involving various RAS components present in intracellular organelles exists in neurons, with both AT_1_ and AT_2_ receptors observed in the endoplasmic reticulum [101,103]. This intracellular RAS may modulate the effects of the extracellular RAS in dopaminergic neurons [62,103].

It is noteworthy that in the subgroup of pyramidal cells that did not respond with potentiation to Ang II, at higher Ang II concentrations (1–10 micromolar), a slight tendency toward inhibition was observed due to the appearance of individual cells showing inhibitory responses. The physiological significance of this occurrence remains unknown. Since neither AT_1_ nor AT_2_ antagonists influenced this phenomenon, we hypothesize that Ang II can be converted to shorter angiotensin sequences, such as Ang III by aminopeptidases A and B, and then further to Ang IV by aminopeptidase B in our rat brain tissue samples [21]. The activation of their specific receptors, such as the AT_4_ receptor, might explain these inhibitory tendencies. This hypothesis may also account for the observation that a lower degree of potentiation was recorded at the highest Ang II concentration (10 μM). It is possible that some responsive cells exhibited a combination of potentiation and inhibition. Alternative possibilities include desensitization of AT_1_ receptors [104,105,106] or disruption of neuronal functioning (neuronal damage) [107,108,109] due to the administration of high concentrations of Ang II. Further research is warranted to elucidate these hypotheses.

A balanced level of glutamate is crucial for normal brain function. However, excessive accumulation or overactivation of its excitatory receptors can induce excitatory signaling and cytotoxicity and ultimately lead to neuronal damage and cell death [56,77,110]. Accordingly, tonic activation of NMDA receptors plays a crucial role in normal neuronal function, but excessive activation can significantly contribute to glutamate-induced excitotoxicity [17,111,112]. In this study, we have elucidated an Ang II–AT_1_ receptor–dopamine D1 receptor-mediated activation of NMDA currents in a subset of layer V pyramidal neurons in the prefrontal cortex. The enhanced excitatory glutamatergic activity may contribute to excitotoxicity and neurodegeneration, consistent with the existing literature.

Our study was conducted using brain slices from young rats rather than mature rats. Consequently, the applicability of our findings to older age groups should be interpreted with caution. Nonetheless, the existing scientific literature indicates that the expression of AT_1_ receptors in most brain regions of adult rats is comparable to or even exceeds that observed in young rats [43]. Further research is needed to elucidate the relevance of these findings to older animals. Additionally, caution is required when extrapolating our results to human (patho)physiology. Human protein databases indicate that the expression of AT_1_ receptor mRNA in the human prefrontal cortex is very low [113]. However, a recent study has revealed that AT_1_ receptor gene expression increases with aging in humans. Advanced stages of Alzheimer’s disease and mixed dementias have also been associated with elevated AT_1_ receptor expression in the frontal cortex [114]. Although AT_1_ receptor gene variants are well established as being associated with essential hypertension [115], evidence linking these variants to human neuropsychiatric disorders remains inconclusive. Nevertheless, associations have been observed between AT_1_ receptor gene variants and Parkinson’s disease, Alzheimer’s disease, and other forms of dementia [116,117]. These findings align with reports suggesting that ligands targeting RAS components may have potential therapeutic effects in neuropsychiatric conditions [18,21,22,58,59,64]. Our data reveal an important mechanism in RAS-related prefrontal pathophysiology, enhancing our understanding of the impact of RAS elements on neuronal functions. This could contribute to elucidating and refining the potential therapeutic benefits of ligands targeting RAS components in pathological conditions.

## 4. Materials and Methods

### 4.1. Brain Slice Preparation

Brain slices from young Wistar rats (9–12 days old) were prepared following the guidelines and with the approval of the Ethical Board of Semmelweis University, Budapest, Hungary, in accordance with the Declaration of the European Communities Council Directives. The rats were decapitated, and their brains were rapidly extracted and immersed in ice-cold artificial cerebrospinal fluid (aCSF). The aCSF was saturated with 95% oxygen and 5% carbon dioxide and contained the following components (in mM): NaCl 126, KCl 2.5, NaH_2_PO_4_ 1.2, CaCl_2_ 2.4, MgCl_2_ 1.3, NaHCO_3_ 26, and glucose 10, adjusted to pH 7.4. Thin slices, 200 μm thick, were prepared from hemisected forebrains using a tissue slicer (MA752, Campden Instruments, Kensington, UK), specifically targeting the prefrontal area of the neocortex. After sectioning, 6–8 slices from a single brain were placed in a holding chamber and maintained in oxygenated aCSF at 36 °C for at least 1 h before use. Subsequently, individual slices were placed into the recording chamber with a volume of 300–400 μL and continuously superfused at a rate of 2.5–3 mL per minute with oxygenated aCSF at room temperature (22–25 °C). The composition of the aCSF used for superfusion remained consistent with that used for incubation, except when studying the effects of a Ca^2+^-free environment. In this scenario, CaCl_2_ was omitted, while the remaining composition of the aCSF remained unchanged. Each slice was provided with a minimum recovery period of 15 min prior to the initiation of each experiment.

### 4.2. Tight Seal Whole-Cell Recording

To measure membrane currents of pyramidal cells in layer V of the PFC in brain slices, we employed procedures similar to those described in prior studies, as outlined by Edwards et al. [118]. Pyramidal cells in layer V of the PFC were visualized using an upright interference contrast microscope equipped with a 40× water immersion objective (Axioskop 2 FS; Carl Zeiss, Baden-Württemberg, Germany). Patch pipettes were pulled using a micropipette puller (Narishige PP-83, Narishige, Tokyo, Japan) from borosilicate glass capillaries with an outer diameter of 2 mm. The patch pipettes were filled with an intracellular solution containing the following components (in mM): K-gluconate 140, NaCl 10, MgCl_2_ 1, HEPES (N-(2-hydroxyethyl)piperazine-N′-(2-ethanesulfonic acid)) 10, EGTA (ethylene glycol-bis-(β-aminoethylether)-N,N,N′,N′-tetraacetic acid) 11, MgATP 1.5, and LiGTP 0.3. The pH was adjusted to 7.3 with KOH. The resistance of the pipettes ranged between 3 and 7 megaohms (MΩ). The liquid junction potential (V_LJ_) between the bath and pipette solutions at 22 °C was determined following the method according to Barry [119], resulting in a value of 15.2 mV. All membrane potential values reported in this study were adjusted to account for the V_LJ_. Pyramidal cell membrane potentials were recorded using the current clamp mode of the patch-clamp amplifier (Axopatch 200B, Molecular Devices, San Jose, CA, USA) immediately upon establishing whole-cell configuration. It was typically within the range of −60 to −80 mV. Subsequently, the system was allowed to stabilize for 5–10 min to achieve equilibrium between the patch pipette and the intracellular environment. Following this, currents induced by NMDA were measured in the voltage-clamp mode. Whole-cell recordings, lasting 40–50 min each, were consistently achieved, with cells exhibiting stable membrane properties throughout. The data were filtered at 2 kHz using the built-in filter of the Axopatch 200B, digitized at 5 kHz, and stored on a laboratory computer using a Digidata 1200 interface and pClamp 8.0 software (Molecular Devices).

### 4.3. Application of Drugs

Drugs were administered by altering the superfusion medium using three-way valves. With a constant flow rate of 2.5–3 mL/min, it took approximately 20 s for the drugs to reach the bath.

### 4.4. Immunohistochemistry

Whole brains of 10-day-old Wistar rats were prepared as described by Gage et al. [120], with the modification that 10% neutral buffered formalin (Leica Biosystems, Nußloch, Germany) was used as the fixative. After postfixing in 10% neutral buffered formalin for 24 h, the brains were embedded in EM-400 paraffin (Leica Biosystems, Germany). Tissue sections of 4 μm thickness were cut using an RM2245 Semi-Automated Rotating Microtome (Leica Biosystems, Germany) and mounted on SuperFrost Plus adhesion slides (Thermo Fisher Scientific, Waltham, MA, USA). After deparaffinization and rehydration, heat-induced epitope retrieval was performed using a 100-fold diluted citric acid-based antigen unmasking solution (Vector Laboratories, Newark, CA, USA). Endogenous peroxidase activity was inhibited with 3% H_2_O_2_. Following overnight incubation at 4 °C in blocking buffer (phosphate-buffered saline (PBS), pH 7.4, containing 2.5% bovine serum albumin, 2.5% non-fat milk, 2.5% normal goat serum, and 2.5% normal horse serum), sections were stained with 6.5 µg/mL anti-AT1R rabbit polyclonal antibody (MBS151548, MyBioSource, San Diego, CA, USA) diluted in the blocking buffer for 1 h at room temperature (RT). To assess antibody specificity, parallel samples were stained with the same antibody preincubated with an equivalent quantity of the MBS152017 AT1R blocking peptide (MyBioSource, USA) for 1 h at RT. Immunoreactivity for AGTR1 was detected with subsequent incubations using SignalStain^®^ Boost IHC Detection Reagent (HRP, rabbit) (8114, Cell Signaling Technology, Danvers, MA, USA) and a TSA Plus Fluorescein Kit (Perkin Elmer, USA). All sections were counterstained with 4′,6-diamidino-2-phenylindole, dihydrochloride (D1306, Thermo Fisher Scientific, USA) and mounted in a Fluoromount-G™ Mounting Medium (Thermo Fisher Scientific, USA). Specimens were examined using a Leica SP8 Confocal Microscope (Leica Biosystems, Germany).

### 4.5. In Situ Hybridization

A 610 bp long fragment of the rat AT1R cDNA (Genbank accession: BC078810) was subcloned into a pBluescript II SK vector and used as a template for in vitro transcription of FITC-12-UTP (Merck, Budapest, Hungary)-labeled probes, according to the MAXIScript T3/T7 KIT (Thermo Fisher Scientific, Budapest, Hungary). The perfused–fixed brains of 10-day-old mice were sectioned at 12 μm thickness using a cryostat (Leica Microsystems GmbH, Wetzlar, Germany). The sections were subsequently thaw-mounted and air-dried at 37 °C on positively charged Superfrost Plus slides (Thermo Fisher Scientific) and stored at −80 °C. Hybridizations were conducted as previously described [121], with the exception that, as an initial step, the sections were treated with 500 µg/mL pepsin diluted in 0.2 M HCl for six minutes at 37 °C, and then rinsed three times in DEPC-treated water (all from Merck). The slides were incubated overnight in humid chambers at 55 °C with 400 ng/μL of the FITC 12-UTP-labeled probes. On the following day, the sections were washed, incubated in 5% normal goat serum containing 0.5% TX-100 in PBS (Merck) for 15 min and then incubated overnight at 4 °C with HRP-conjugated goat anti-FITC antibody (1:200, Thermo Fisher). The signal was then visualized using the TSA Fluorescein System (Perkin Elmer, Perform Hungaria Kft, Budapest, Hungary). Images were acquired using a 780LSM confocal laser-scanning microscope (Carl Zeiss Technika Kft., Budapest, Hungary).

### 4.6. Materials

The following drugs and chemicals were used: eprosartan mesylate, N-Methyl-D-aspartic acid (NMDA), PD 123319 di(trifluoroacetate) salt hydrate, (±)-sulpiride (Sigma-Aldrich, St. Louis, MO, USA), angiotensin II, SCH 23390 hydrochloride, and tetrodotoxin citrate (Tocris). Stock solutions (10–100 mM) were made using distilled water or dimethylsulfoxide (DMSO) and were subsequently diluted in the medium (daily, using either the extracellular or intracellular solution as appropriate). The final concentration of DMSO was kept below 0.1% and had no impact when used independently.

### 4.7. Statistics

Multiple comparisons with the control value were performed using a one-way ANOVA followed by Bonferroni’s post hoc test. A probability level of 0.05 or less was considered statistically significant. Means ± SEM are presented throughout.

## Figures and Tables

**Figure 1 ijms-25-12644-f001:**
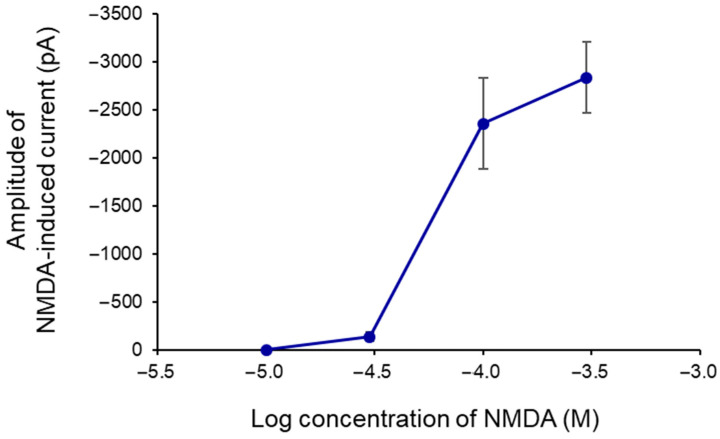
NMDA-evoked inward currents in layer V pyramidal neurons of the rat prefrontal cortex. Whole-cell patch-clamp measurements at a holding potential of −70 mV. Correlation of concentration–response amplitudes for NMDA-induced currents. Each data point corresponds to measurements taken from *n* cells for each NMDA concentration (10 µM, *n* = 3; 30 µM, *n* = 6; 100 µM, *n* = 7; 300 µM, *n* = 6).

**Figure 2 ijms-25-12644-f002:**
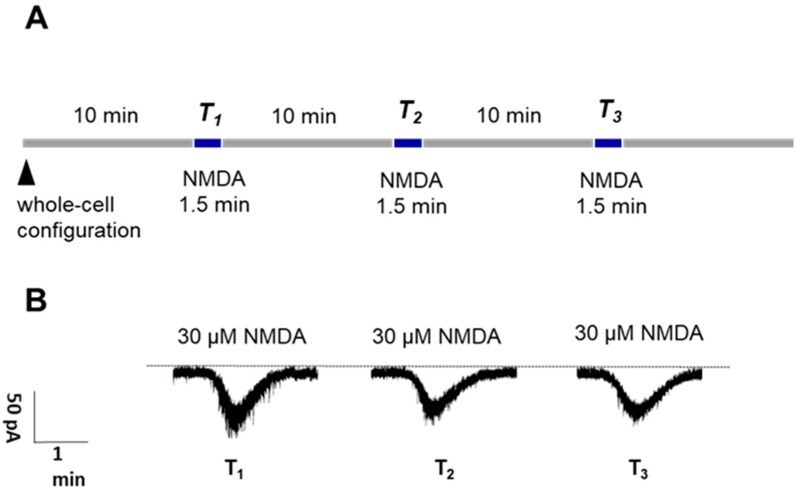
30 µM NMDA-induced inward currents in layer V pyramidal neurons of the rat PFC. (**A**) Diagram of the experimental patch-clamp protocol. After the whole-cell configuration was established, 200 µm thick mPFC slices were superfused with aCSF for 10 min to achieve diffusion balance between the patch pipette and the cell interior. Then, 30 μM NMDA was applied three times for 1.5 min (T_1_, T_2_, T_3_), separated by superfusion periods of 10 min with drug-free aCSF. The membrane currents were measured using the amplifier in voltage-clamp mode at a holding potential of −70 mV. The amplitude of the NMDA-induced ion currents was quantified. (**B**) Representative tracing of the current response to NMDA after three applications of 30 μM NMDA (T_1_, T_2_, T_3_).

**Figure 3 ijms-25-12644-f003:**
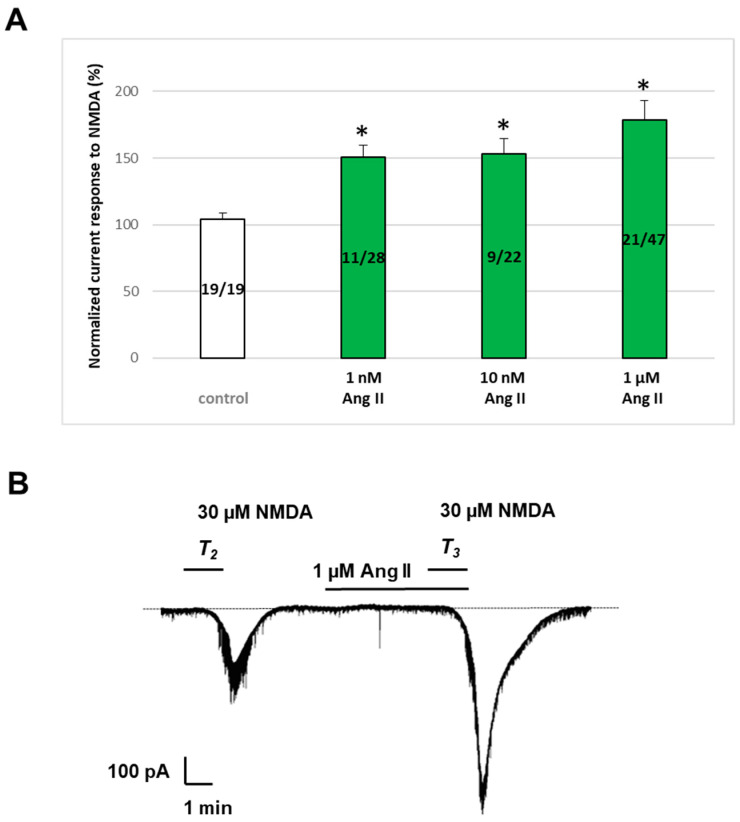
Effect of 1 nM–1 µM Ang II on NMDA-induced inward currents in layer V pyramidal neurons of the rat prefrontal cortex. Whole-cell patch-clamp measurements were conducted at a holding potential of −70 mV. A total of 30 µM of NMDA was applied three times for 1.5 min (T_1_, T_2_, T_3_) with a 10 min interval between applications. Under these conditions, current responses were consistent at T_2_ and T_3_. Ang II at concentrations of 1 nM–1 µM was applied for 5 min before and during T_3_. (**A**) Mean ± SEM of *n* experiments, showing the effects of 0 µM (control) (*n* = 19), 1 nM (*n* = 11/28), 10 nM (*n* = 9/22), and 1 µM (*n* = 21/47) Ang II on NMDA currents with respect to the response measured at T2. Green bars represent the normalized NMDA-induced current responses (%) in the subpopulation of mPFC layer V pyramidal cells in which Ang II potentiated ion currents. * *p* < 0.05, a significant difference from the normalized current responses to NMDA under control conditions. (**B**) Representative tracing of a current response to 30 μM NMDA after T_2_ and T_3_, and in the presence of 1 µM Ang II for 5 min before and during T_3_.

**Figure 4 ijms-25-12644-f004:**
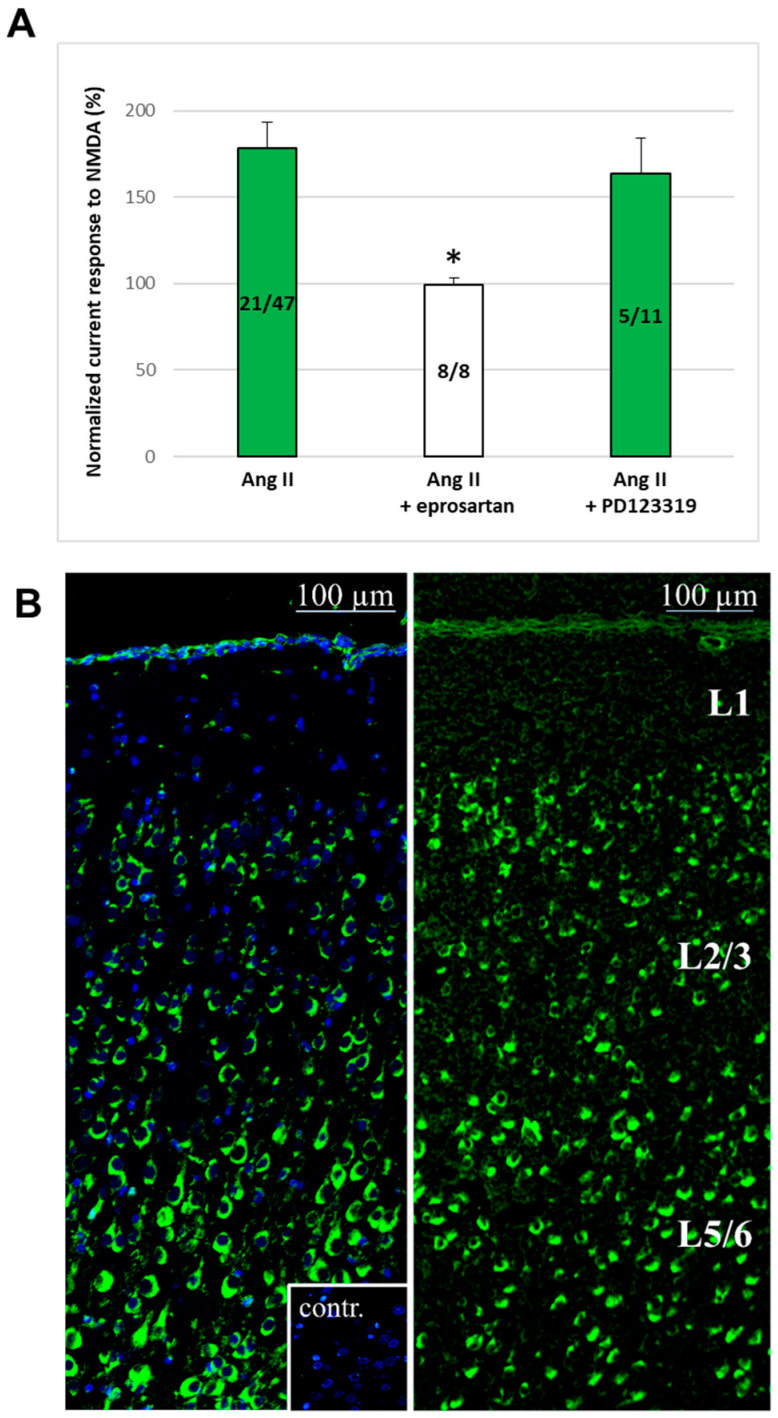
Role of AT_1_ and AT_2_ receptors in the potentiation of NMDA receptor function by Ang II and protein/mRNA expression of AT_1_ receptors (AT1R) in the rat mPFC. (**A**) NMDA-induced current responses in mPFC layer V pyramidal cells were detected using whole-cell patch-clamp measurements at a holding potential of −70 mV. A 30 µM concentration of NMDA was administered three times for 1.5 min each (T_1_, T_2_, T_3_) with a 10 min interval between applications. A 1 µM concentration of Ang II was applied for 5 min before and during T_3_. In separate experiments, aCSF contained either 1 µM eprosartan (AT_1_ antagonist) or 5 µM PD 123319 (AT_2_ antagonist) throughout the entire measurement period. The data are presented as the mean ± SEM of *n* experiments: effects of 1 µM Ang II (*n* = 21/47), 1 µM Ang II + 1 µM eprosartan (*n* = 8), and 1 µM Ang II + 5 µM PD 123319 (*n* = 5/11) on NMDA currents at T_3_, normalized with respect to the response measured at T_2_. Green bars indicate normalized NMDA-induced current responses (T_3_/T_2_ %) in cells where Ang II potentiated NMDA receptor-mediated ion currents. If potentiation cannot be observed, all cells in the group are represented. * *p* < 0.05 indicates a significant difference from the 1 µM Ang II potentiation group. (**B**) Immunohistochemical detection of AT_1_ receptor protein expression (**left** panel) was conducted using the MBS151548 anti-AT1R rabbit polyclonal antibody, and fluorescent in situ hybridization analysis of AT1R mRNA expression (**right** panel) was performed in the mPFC of 10-day-old Wistar rats. For immunohistochemical detection, the MBS151548 anti-AT1R rabbit polyclonal antibody preincubated with the MBS152017 AT1R blocking peptide served as the negative control (**left** panel, square below). Blue indicates cells counterstained with 4′,6-diamidino-2-phenylindole (DAPI), green indicates cells stained with anti-AT1R rabbit polyclonal antibody (left panel), and cells expressing AT1R mRNA (**right** panel). L1, L2/3, and L5/6 denote the layers of the mPFC.

**Figure 5 ijms-25-12644-f005:**
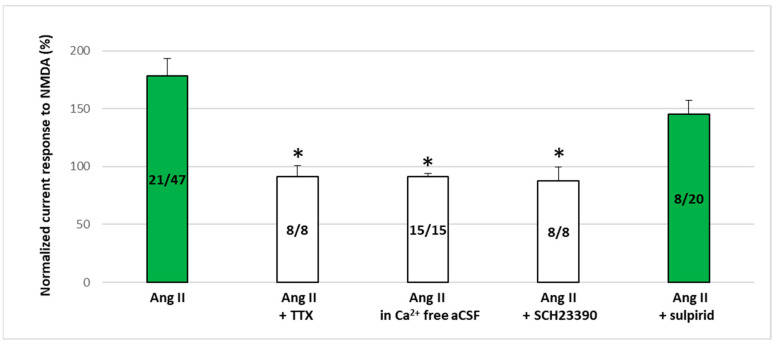
The effect of synaptic isolation and the role of D1 and D2 dopaminergic receptors in Ang II-induced NMDA receptor potentiation. NMDA-induced current responses in mPFC layer V pyramidal cells were detected using whole-cell patch-clamp measurements at a holding potential of −70 mV. A total of 30 µM of NMDA was administered three times for 1.5 min (T_1_, T_2_, T_3_) with a 10 min interval between applications. A total of 1 µM of Ang II was applied for 5 min before and during T_3_. To test the effect of synaptic isolation, 0.5 µM tetrodotoxin (TTX) was added to the aCSF, or Ca^2+^-free aCSF was used throughout the entire experiment. To detect D1 receptor and D2 receptor involvement in Ang II-induced potentiation of the NMDA receptor, 10 µM SCH-23390 (D1 receptor antagonist) or 20 µM sulpiride (D2 receptor antagonist) was added to the aCSF throughout the measurements. Mean ± SEM of *n* experiments: effects of 1 µM Ang II (*n* = 21/47), 1 µM Ang II + 0.5 µM TTX (*n* = 8), 1 µM Ang II + Ca^2+^-free aCSF (*n* = 15), 1 µM Ang II + 10 µM SCH-23390 (*n* = 8), and 1 µM Ang II + 20 µM sulpiride (*n* = 8/20) on NMDA currents at T_3_ normalized with respect to the response measured in T_2_. Green bars indicate normalized NMDA-induced current responses (T_3_/T_2_ %) in cells where Ang II potentiated NMDA receptor-mediated ion currents. If potentiation cannot be observed, all cells in the group are represented. * *p* < 0.05 indicates a significant difference from the 1 µM Ang II potentiation group.

## Data Availability

The data that support the findings of this study are available from the corresponding author upon reasonable request.

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
