# Peer review of "Potentiation of NMDA Receptors by AT1 Angiotensin Receptor Activation in Layer V Pyramidal Neurons of the Rat Prefrontal Cortex†"

_ijms, 2024, doi:10.3390/ijms252312644_

Round 1
Reviewer 1 Report (New Reviewer)
Comments and Suggestions for Authors
The manuscript by Hanuska et al. explores the critical role of angiotensin II in the central nervous system, specifically its influence on glutamatergic responses in regions associated with cognition and learning (layerV/VI). The authors, experienced in the methodologies employed, address an important topic. However, several key issues need to be addressed before a final decision can be made on the manuscript.
Here are my primary concerns:
1. Introduction Clarity: The introduction lacks sufficient detail regarding the role of angiotensin II in layer V/VI pyramidal neurons, particularly its origin. It is essential to clarify where angiotensin II is synthesized, how it is released within layer V/VI neurons, and the role of astrocytes in the formation and origin of angiotensinogen.
2. Cell Response Limitations: The study finds that only 40% of the cells in layer V/VI respond to angiotensin II, yet the authors focus solely on these responsive cells. This raises concerns about the overall significance of the findings, as the majority of cells do not exhibit a response. It suggests that the relevance of the angiotensin II mechanism may be overstated given the limited subset of cells selected for investigation.
3. Inclusion of Preliminary Data: The manuscript currently includes preliminary data that are not fully developed, specifically the statement: "Our preliminary data indicate that application of angiotensin IV (100 nM) resulted in inhibitory responses in 3 of 7 pyramidal cells, with a T3/T2 ratio of 0.575±0.06." This type of data should be removed, as it does not contribute to the manuscript’s findings.
4. Concentration Relevance: The concentrations of angiotensin II used in the experiments appear to be high. It is crucial to clarify whether these concentrations are physiological or supraphysiological.
5. AT1 Receptor Localization: In Figure 4, while the presence of AT1 receptors in layer V/VI is demonstrated, the manuscript does not adequately explain their specific localization or the origin of angiotensin II. The authors should provide a detailed description of angiotensin II's origin and the precursors involved in its formation in layer V/VI.
6. Conclusion Specificity: The conclusion should specify which subpopulations of cells in layer V/VI respond to angiotensin II. The definition and characterization of this subset need to be clearer, especially considering that responsive cells represent a minority within the layer.
Addressing these concerns will strengthen the manuscript and clarify its contributions to the field.
Author Response
Thank you very much for your thorough and supportive review.
In response to your questions, suggestions, and concerns regarding the manuscript, we have made the following revisions. All changes are highlighted in blue text within our responses and in the manuscript.
„Introduction Clarity: The introduction lacks sufficient detail regarding the role of angiotensin II in layer V/VI pyramidal neurons, particularly its origin. It is essential to clarify where angiotensin II is synthesized, how it is released within layer V/VI neurons, and the role of astrocytes in the formation and origin of angiotensinogen.”
We have added a more detailed description of Ang II synthesis in the introduction, including the role of astrocytes, supported by additional references.
„Cell Response Limitations: The study finds that only 40% of the cells in layer V/VI respond to angiotensin II, yet the authors focus solely on these responsive cells. This raises concerns about the overall significance of the findings, as the majority of cells do not exhibit a response. It suggests that the relevance of the angiotensin II mechanism may be overstated given the limited subset of cells selected for investigation.”
We have expanded the discussion to include a more detailed description of the issue of cell response limitations, supported by additional references.
„Inclusion of Preliminary Data: The manuscript currently includes preliminary data that are not fully developed, specifically the statement: "Our preliminary data indicate that application of angiotensin IV (100 nM) resulted in inhibitory responses in 3 of 7 pyramidal cells, with a T3/T2 ratio of 0.575±0.06." This type of data should be removed, as it does not contribute to the manuscript’s findings.”
We have removed these data.
„Concentration Relevance: The concentrations of angiotensin II used in the experiments appear to be high. It is crucial to clarify whether these concentrations are physiological or supraphysiological.”
We have expanded the discussion to clarify whether the Ang II concentrations used in our study were physiological or supraphysiological, with support from additional references.
„AT1 Receptor Localization: In Figure 4, while the presence of AT1 receptors in layer V/VI is demonstrated, the manuscript does not adequately explain their specific localization or the origin of angiotensin II. The authors should provide a detailed description of angiotensin II's origin and the precursors involved in its formation in layer V/VI.”
We have expanded the discussion to include a description of the origin of Ang II, its precursors, and the potential involvement of various cell types in its formation in the cortex, supported by additional references.
„Conclusion Specificity: The conclusion should specify which subpopulations of cells in layer V/VI respond to angiotensin II. The definition and characterization of this subset need to be clearer, especially considering that responsive cells represent a minority within the layer.”
We have expanded the discussion by comparing our findings with relevant scientific publications to suggest which cell populations may be involved in the observed Ang II-NMDA interaction, supported by additional references. We have also contextualized our data within the broader sequence of scientific advancements in this field, highlighting its significance.

Reviewer 2 Report (New Reviewer)
Comments and Suggestions for Authors
The publication provides interesting evidence that angiotensin II (Ang II) can modulate NMDA receptors in layer V pyramidal neurons of the prefrontal cortex (PFC), which is important for understanding cognitive and emotional mechanisms. The authors performed detailed electrophysiological experiments using the patch-clamp technique to demonstrate the effect of Ang II on NMDA-induced current potentials, and also investigated the role of AT1 and AT2 receptors. Despite the well-designed experiment, the work could have benefited from a more detailed analysis of the interactions between Ang II and dopamine receptors. I have concerns regarding the fact that SCH-23390 (a D1 receptor antagonist) also inhibits GIRK (G protein-coupled inwardly rectifying potassium) channels, which may be relevant in the context of this study. If SCH-23390 blocks GIRK channels, the effects attributed to D1 receptor antagonism could be confounded by GIRK inhibition, as GIRK channels also modulate neuronal excitability. Since GIRK channels influence membrane potential, their inhibition could alter neuronal responses to NMDA and Ang II, potentially obscuring the specific role of D1 receptors in cognitive functions. Could the authors address these concerns? Thank you.
Author Response
Thank you very much for your thorough and supportive review.
In response to your questions, suggestions, and concerns regarding the manuscript, we have made the following revisions. All changes are highlighted in blue text within our responses and in the manuscript.
„Despite the well-designed experiment, the work could have benefited from a more detailed analysis of the interactions between Ang II and dopamine receptors.”
We have expanded the discussion to include aspects of the interactions between Ang II and dopamine receptors. We have also contextualized our data within the broader sequence of scientific advancements in this field, highlighting its significance.
„I have concerns regarding the fact that SCH-23390 (a D1 receptor antagonist) also inhibits GIRK (G protein-coupled inwardly rectifying potassium) channels, which may be relevant in the context of this study. If SCH-23390 blocks GIRK channels, the effects attributed to D1 receptor antagonism could be confounded by GIRK inhibition, as GIRK channels also modulate neuronal excitability. Since GIRK channels influence membrane potential, their inhibition could alter neuronal responses to NMDA and Ang II, potentially obscuring the specific role of D1 receptors in cognitive functions. Could the authors address these concerns?”
We have expanded the discussion to include the potential consequences and significance of the GIRK inhibitory action of SCH-23390 in the context of our experiments.

Round 2
Reviewer 1 Report (New Reviewer)
Comments and Suggestions for Authors
The authors have addressed my comments thoroughly and thoughtfully
This manuscript is a resubmission of an earlier submission. The following is a list of the peer review reports and author responses from that submission.
Round 1
Reviewer 1 Report
Comments and Suggestions for Authors
The paper tested the potential of NMDA receptors in rats with whole patch clamp experiments, followed by a series of antagonist tests to verify the results. The experiment itself is well-founded and logical. However, there are still some minor problems that can improve the general quality of the paper if revised:
1. The description in the introduction, with the sentence “Both receptors are expressed in the brain, with relatively moderate densities in cortical areas.” Sounds vague to the readers. How moderate is the densities in such area comparing to other ATs? The authors may need to add more references. Also, the description of “signaling cascade that induce long-term modifications” triggered by NMDA receptors meets the same problem.
2. In the results, the “slightly decrease from the first to second application” is not the reason for the lack of analysis. The authors should focus on both evaluation for T1 to T2, and T2 to T3 equally. Thus, the readers can see the comparison with no bias.
3. The normalized current response to NMDA in responsive cells at 1nM and 10nM, although both groups have significance with the control group, there seems no significance compared to each other while the concentration of the cells is 10x. Is there any explanation for this result? Also, if the authors can add one more group (e.g., at 10μM) for comparison it would be much clearer to the readers if there would be any detrimental effects to the cells.
4. In results 2.3., the effect of eprosartan (1 μM) and PD123329 (5 μM) were not presented. Why?
5. In the discussion section, authors have indicated the observation of a slight tendency towards inhibition for Ang II, while there’s lack of explanation of the mechanism or any hypothesis for this observation. Please add more explanation to support this discovery, and how it might connect to the NMDA receptors throughout the pathways logically.
Author Response
Thank you very much for your thorough and supportive review.
In response to your questions, advice, and concerns about the manuscript, we have made the following corrections to the MS. Changes are indicated with blue text in our responses and in the MS. Green text in the MS indicates changes with unaltered meaning in response to the duplicate issue with my previous publications, as pointed out by the editors.
„The description in the introduction, with the sentence “Both receptors are expressed in the brain, with relatively moderate densities in cortical areas.” Sounds vague to the readers. How moderate is the densities in such area comparing to other ATs? The authors may need to add more references.”
We have included a more detailed description of AT1 and AT2 receptor expression, supported by additional references, in the introduction.
„Also, the description of “signaling cascade that induce long-term modifications” triggered by NMDA receptors meets the same problem.”
We have provided a more detailed description of the involvement of NMDA receptors in synaptic plasticity in the introduction.
„In the results, the “slightly decrease from the first to second application” is not the reason for the lack of analysis. The authors should focus on both evaluation for T1 to T2, and T2 to T3 equally. Thus, the readers can see the comparison with no bias.”
This phenomenon has been extensively observed and described in our previous studies, to which we refer and cite as references.
„The normalized current response to NMDA in responsive cells at 1nM and 10nM, although both groups have significance with the control group, there seems no significance compared to each other while the concentration of the cells is 10x. Is there any explanation for this result? Also, if the authors can add one more group (e.g., at 10μM) for comparison it would be much clearer to the readers if there would be any detrimental effects to the cells.”
We included data on 10 micromolar AngII for comparison. Additionally, we expanded the discussion with brief texts addressing these issues.
„In results 2.3., the effect of eprosartan (1 μM) and PD123329 (5 μM) were not presented. Why?”
We added these data to the text.
„In the discussion section, authors have indicated the observation of a slight tendency towards inhibition for Ang II, while there’s lack of explanation of the mechanism or any hypothesis for this observation. Please add more explanation to support this discovery, and how it might connect to the NMDA receptors throughout the pathways logically.”
In relation to this issue, we included preliminary data on the effects of angiotensin IV in the results section (lines 164-166) and provided a more detailed discussion of this phenomenon.

Reviewer 2 Report
Comments and Suggestions for Authors
The authors provide clear evidence that Ang II potentiates 9-12 day old Wistar rat pyramidal neuron NMDA receptor currents in the presence of physiological level of extracellular magnesium and in the absence of added NMDA co-agonists. Use of selective pharmacological tool compounds allow the authors to conclude that angiotensin 1 (AT1) receptor stimulation by Ang II is likely to mediate NMDA current potentiation. Further, omission of extracellular calcium as well as 500 nM TTX, a blocker of voltage-gated sodium channels, abolished Ang II-mediated NMDA potentiation suggesting that Ang II regulates presynaptic AT1 receptors. Interestingly, selective dopamine D1 receptor antagonist but not dopamine D2 receptor antagonist was also able to block Ang II-mediated NMDA potentiation. Previous studies have shown that Ang II can elicit dopamine release in the brain and such an effect can be blocked by AT1 receptor antagonists. AT1 receptor protein was visualized from rat brain slices by polyclonal antibody and with AT1 mRNA with fluorescent in situ hybridization.
Clear experimental data obtained from cortical slices support a role for AT1 receptor in the regulation of rat NMDA receptors. My main concern here is how well can we extrapolate these findings to human as there appear to very low (0.1-0.2 nTPM, 0.3 nTPM is thought to be the limit of meaningful mRNA expression) AT1 receptor mRNA (AGTR1 gene) expression in human prefrontal cortex (e.g. the human protein atlas: www.proteinatlas.org) in contrast to clear dopamine D1 receptor mRNA expression (DRD1 gene) in prefrontal cortex. I wonder if the authors could discuss this briefly. Further, absence of genetic evidence for human neurological disease phenotypes associated with AGTR1 gene variants begs the question how relevant AT1R modulation of NMDA receptor function may be in human disease pathology in contrast to essential hypertension, which is genetically linked with AGTR1, please discuss.
Minor point:
9-12 day old rats are quite young in terms of neuronal circuit maturation. It is well known known that brain slices from more mature rats (older than 1 month) are less viable and harder to work with. Would you expect similar results & AT1 receptor expression from mature rat brain slices?
Author Response
Thank you very much for your thorough and supportive review.
In response to your questions, advice, and concerns about the manuscript, we have made the following corrections to the MS. Changes are indicated with blue text in our responses and in the MS. Green text in the MS indicates changes with unaltered meaning in response to the duplicate issue with my previous publications, as pointed out by the editors.
„My main concern here is how well can we extrapolate these findings to human as there appear to very low (0.1-0.2 nTPM, 0.3 nTPM is thought to be the limit of meaningful mRNA expression) AT1 receptor mRNA (AGTR1 gene) expression in human prefrontal cortex (e.g. the human protein atlas: www.proteinatlas.org) in contrast to clear dopamine D1 receptor mRNA expression (DRD1 gene) in prefrontal cortex. I wonder if the authors could discuss this briefly. Further, absence of genetic evidence for human neurological disease phenotypes associated with AGTR1 gene variants begs the question how relevant AT1R modulation of NMDA receptor function may be in human disease pathology in contrast to essential hypertension, which is genetically linked with AGTR1, please discuss.”
We added a paragraph at the end of the discussion addressing these issues.
„9-12 day old rats are quite young in terms of neuronal circuit maturation. It is well known known that brain slices from more mature rats (older than 1 month) are less viable and harder to work with. Would you expect similar results & AT1 receptor expression from mature rat brain slices?”
The first two sentences of the newly added paragraph address this question.
